# Cell-Free Supernatant from a Strain of *Bacillus siamensis* Isolated from the Skin Showed a Broad Spectrum of Antimicrobial Activity

**DOI:** 10.3390/microorganisms12040718

**Published:** 2024-04-02

**Authors:** Natalia Pedretti, Ramona Iseppi, Carla Condò, Luca Spaggiari, Patrizia Messi, Eva Pericolini, Alessandro Di Cerbo, Andrea Ardizzoni, Carla Sabia

**Affiliations:** 1Department of Surgical, Medical, Dental and Morphological Sciences with Interest in Transplant, Oncological and Regenerative Medicine, University of Modena and Reggio Emilia, 41125 Modena, Italy; natalia.pedretti@unimore.it (N.P.); eva.pericolini@unimore.it (E.P.); andrea.ardizzoni@unimore.it (A.A.); 2Department of Life Sciences, University of Modena and Reggio Emilia, 41125 Modena, Italy; ramona.iseppi@unimore.it (R.I.); carla.condo@hotmail.it (C.C.); patrizia.messi@unimore.it (P.M.); 3Clinical and Experimental Medicine PhD Program, University of Modena and Reggio Emilia, 41125 Modena, Italy; luca.spaggiari@unimore.it; 4School of Biosciences and Veterinary Medicine, University of Camerino, 62024 Matelica, Italy

**Keywords:** skin microbiota, antibacterial compounds, CFS, *Bacillus siamensis*

## Abstract

In recent years, the search for new compounds with antibacterial activity has drastically increased due to the spread of antibiotic-resistant microorganisms. In this study, we analyzed Cell-Free Supernatant (CFS) from *Bacillus siamensis*, assessing its potential antimicrobial activity against some of the main pathogenic microorganisms of human interest. To achieve this goal, we exploited the natural antagonism of skin-colonizing bacteria and their ability to produce compounds with antimicrobial activity. Biochemical and molecular methods were used to identify 247 strains isolated from the skin. Among these, we found that CFS from a strain of *Bacillus siamensis* (that we named CPAY1) showed significant antimicrobial activity against *Staphylococcus aureus*, *Enterococcus faecalis*, *Streptococcus agalactiae*, and *Candida* spp. In this study, we gathered information on CFS’s antimicrobial activity and on its sensitivity to chemical–physical parameters. Time–kill studies were performed; anti-biofilm activity, antibiotic resistance, and plasmid presence were also investigated. The antimicrobial compounds included in the CFS showed resistance to the proteolytic enzymes and were heat stable. The production of antimicrobial compounds started after 4 h of culture (20 AU/mL). CPAY1 CFS showed antimicrobial activity after 7 h of bacteria co-culture. The anti-biofilm activity of the CPAY1 CFS against all the tested strains was also remarkable. *B. siamensis* CPAY1 did not reveal the presence of a plasmid and showed susceptibility to all the antibiotics tested.

## 1. Introduction

The skin epidermis is a complex environment, and with an average surface area between 1.6 and 2.2 m^2^, it is one of the largest epithelial surfaces of the human body for interaction with microbes. From the early years of life, the skin is colonized by bacteria, fungi, and viruses that make up the skin microbiota. The latter consists of resident microorganisms, mainly behaving as commensals and symbiotes. A stable skin microbiota, characterized by adequate richness and diversity, increases the capacity of the skin to resist colonization by pathogenic bacteria [1]. Microorganisms dwelling on the skin surface, in glandular ducts, and on hair bulbs can be either resident or transient, the latter in the long and short term. The resident microbiota represents a community of microorganisms occurring in appreciable numbers on the skin of most individuals, thanks to their ability to adhere to skin cells. They have adapted to use lipids, urea, and other sweat components as nutrients for their growth. In return, they protect the skin and cooperate with the immune system in counteracting pathogens, by maintaining the local pH (pH ranging from 5.4 to 5.9) and producing secondary metabolites and bacteriocins [2] that inhibit pathogen growth [3]. The skin microbiota also includes transient species coming from the surrounding environment or from the nearby mucosal surfaces [4]. Such species are unable to multiply and colonize the skin surfaces for prolonged periods because they can be easily removed by rubbing or using disinfectants.

The resident microbiota has also been shown to secrete some antimicrobial peptides (AMPs) that work as an effective line of defense. Among these AMPs, bacteriocins are bioactive compounds that have been demonstrated to inhibit microbial pathogens both in vitro and in vivo [5]. Bacteriocins are synthesized at the ribosomal level, and they are usually active against microorganisms phylogenetically related to the producer strains, which are naturally insensitive to them thanks to the development of a specific immunity.

In recent years, the ongoing search for novel compounds with antibacterial activity to stem the spread of antibiotic-resistant microorganisms has sparked increased interest in the protective role of the skin microbiota. Recent discoveries have allowed us to improve our knowledge of the microbiota and of its role in health and disease. This, in turn, has enabled the isolation and characterization of new bioactive compounds/peptides to be employed as novel strategies for controlling and restraining pathogens. As a result, by taking advantage of the natural antagonism of bacteria present on the skin and of their ability to produce compounds with antibacterial activity, the present study aims to investigate and characterize new compounds that can be used as “natural” agents against some of the most important pathogenic microorganisms of human interest [6,7].

Here, we collected microbial samples from the skin of three different body areas. Two hundred and forty-seven bacterial strains were isolated, and their cell-free supernatants (CFSs) were assessed for their antimicrobial activity. According to the screening performed against the main pathogens of human interest, twenty-five “active strains” were found. We focused our attention on a *Bacillus siamensis* (*B. siamensis*) strain, whose identification was obtained by sequencing 16S ribosomal DNA. Indeed, this strain (that we named CPAY1) has been shown to produce compounds capable of exerting antimicrobial activity against a wide range of pathogenic microorganisms, including antibiotic-resistant strains, and has thus been subjected to a preliminary characterization.

## 2. Materials and Methods

### 2.1. Isolation of Microorganisms from Different Skin Sites

Microbiotas were collected by rubbing the skin from four different areas of the body with a swab, from twelve subjects. The following body sites were sampled: the neck and behind the ear (sebaceous sites), the leg (dry site), and the folds of the elbow (wet site). The swabs were put in 1 mL of saline and vortexed, and ten-fold serial dilutions were performed. Then, 100 µL aliquots were seeded onto Tryptic Soy Agar, (TSA; Oxoid S.p.A, Milan, Italy), Mannitol Salt Agar (MSA; Oxoid S.p.A, Milan, Italy) for the isolation of Staphylococci, Yeast Extract Sodium Lactate agar (YELA; Oxoid S.p.A, Milan, Italy) for the isolation of *Propionibacterium* spp., and de Man–Rogosa–Sharpe agar (MRS; Oxoid S.p.A, Milan, Italy) for the isolation of lactic acid bacteria. All plates were incubated aerobically or in anaerobiosis at 37 °C for 48 h. After incubation, colonies were counted and selected according to their different morphologies and colors.

### 2.2. Detection of Antimicrobial Activity of B. siamensis CPAY1

The deferred antagonism assay was performed to preliminarily screen for the presence of antimicrobial activity in two hundred and forty-seven bacterial strains, isolated from skin and cultured in Tryptic Soy Broth (TSB; Oxoid S.p.A, Milan, Italy) at 37 °C for 24 h [8]. The indicator microorganisms (Gram-positive, Gram-negative, and fungal pathogens that most frequently cause community-acquired and nosocomial infections) (see Table 1) were cultured in TSB for 24 h, at incubation temperatures of 30 °C or 37 °C, depending on the species. The strain showing the widest inhibition halo around the spot was then inoculated into fresh TSB, grown overnight at 37 °C, and stocked in 20% glycerol at −80 °C for further characterization.

Furthermore, the CFS from the strain with the best antimicrobial activity, i.e., *B. siamensis* CPAY1, was tested by agar well diffusion assay [9] against the microorganisms found to be particularly susceptible in the initial screening. CFSs were obtained from 10 mL overnight bacterial cultures by centrifugation (10,000× *g* for 20 min at 4 °C), followed by filtration with a 0.45 µm-pore-size filter (Millipore Corp., Bedford, MA, USA). The potential bacterial contamination of CFSs was excluded by incubating 1 mL of each CFS at 37 °C and checking the turbidity (from 24 h to 72 h). In the well agar diffusion method, the negative control is given by the TSB.

### 2.3. Kinetics of Growth and CPAY1 CFS Biosynthesis

TSB (250 mL) was inoculated with 100 μL of *B. siamensis* CPAY1 for 18 h at 37 °C. At appropriate intervals, CPAY1 CFS was collected for the measurement of optical density. The absorbance was measured spectrophotometrically at 630 nm (OD630) by a Sunrise Microplate Reader (Sunrise Tecan, Grödig, Austria); pH values of cultures were also evaluated. The antibacterial activity was evaluated by assaying CPAY1 CFS serial two-fold dilutions against *S. aureus* ATCC 6538. The antimicrobial titer was defined as the reciprocal of the highest dilution producing a distinct inhibition of the indicator lawn and expressed in terms of arbitrary units per milliliter (AU/mL) [10]. The experiments were repeated three times.

### 2.4. CPAY1 CFS Heat Stability, and Protease and pH Sensitivity

To evaluate its heat stability, CPAY1 CFS was incubated for 10 min at 37, 60, 70, 80, 90, and 100 °C, and for 15 min at 121 °C. The sensitivity of CPAY1 CFS to enzymatic degradation by catalase and proteolytic enzymes was evaluated using catalase at pH 7.0 (50 mM potassium phosphate buffer); trypsin, pepsin, and proteinase K at pH 7.5 (100 mM Tris-HCl buffer); and using pepsin at pH 3.0 (50 mM glycine buffer added to 20 mM CaCl_2_) (all enzymes were purchased from Sigma-Aldrich, Milan, Italy). Aliquots of the CFSs at different pH values (2.0, 4.0, 8.0, and 10.0) and their respective controls were incubated (1:1 *v*/*v*) with enzyme solutions (1 mg/mL) at 37 °C for 2 h [11]. Inhibitory activity was determined for retentate and ultrafiltrate against *S. aureus* ATCC 6538 as an indicator microorganism. A pilot experiment had been carried out to exclude a possible antimicrobial role of TSB at different pHs. To test the stability during refrigerated storage, CFS samples, stored at 4 °C for up to 6 months, were collected at different time intervals, and the residual antibacterial activity was determined. The experiments were repeated three times.

### 2.5. Time–Kill Studies

The growth of all the test strains was determined by calculating the change in the optical density of cells grown in contact with the CPAY1 CFS. In a 96-well sterile microplate, 100 μL of sterile nutrient broth was mixed with 50 μL of CPAY1 CFS and with 50 μL of each microbial strain; the latter was thawed from the respective stocks and previously diluted to obtain a cell density of approximately 10^5^ CFU/mL. After orbital shaking at 150 rpm for a total of 26 h, measurements were performed at 1 h intervals at the optical density (OD) of 595 nm by using an automatic microplate reader. The experiments were repeated three times.

### 2.6. Anti-Biofilm Activity

The effect of CPAY1 CFS on the 24 h biofilms from bacterial strains, also used as indicators in the agar well diffusion assay, was tested. These strains were selected because of their antibiotic resistance: MRSA and vancomycin-resistant *E. faecalis*. They were allowed to form on 96-well polystyrene microtiter plates, using approximately 10^5^ CFU/mL of each microbial strain; the plates were incubated at 37 °C. After biofilm formation, the medium was gently aspirated, and plates were washed three times with a sterile phosphate-buffered saline solution (PBS, pH 7.2) to remove planktonic bacteria. Then, CPAY1 CFS was added at 20 AU/mL. Following an additional incubation for 24 h at 37 °C, the biofilm biomass was quantified using the crystal violet staining method [12]. Absorbance values were measured at 570 nm using a microtiter plate reader. The experiments were repeated three times.

### 2.7. The 16S rRNA Sequencing

Genomic DNA of *B. siamensis* CPAY1 was extracted using PrepMan™ Ultra Sample Preparation Reagent (ThermoFisher SCIENTIFIC, Milan, Italy). Sequencing was conducted by MicroSEQ^TM^ 500 16S rDNA sequencing kit (ThermoFisher SCIENTIFIC, Milan, Italy) and analyses of the sequencing data were performed utilizing MicroSEQ ID software (v4.0, ThermoFisher SCIENTIFIC, Milan, Italy).

### 2.8. Antibiotic Resistance

Antibiotic Susceptibility Testing (AST) was performed on the *B. siamensis* CPAY1 strain against gentamicin, clindamycin, erythromycin, kanamycin, tetracycline, streptomycin, and vancomycin by using Etest^®^ Strips (AB Biodisk, Solna, Sweden). The minimum inhibitory concentration (MIC) of the seven antibiotics was determined according to the 2015 Clinical Laboratory Standards Institute (CLSI) guidelines [13]. Resistance and sensitivity to antibiotics were defined using the European Food Safety Authority’s (EFSA’s) breakpoints [14].

### 2.9. Plasmid Isolation

Mini-scale extraction of plasmids from *B. siamensis* CPAY1 was performed according to the O’Sullivan and Klaenhammer method [15]. Agarose gel electrophoresis was carried out on 0.7% agarose gel (5-75182; Sigma-Aldrich, Milan, Italy) in TBE buffer (Tris–Borate–EDTA; Sigma-Aldrich, Milan, Italy) at 60 V for 4 h. Plasmid sizes were estimated by using *E. coli* V517, which contains plasmids of known sizes [16].

### 2.10. Statistical Analysis

The statistical analysis was performed using the GraphPad Prism 9.2.0. (San Diego, CA, USA) program. The significance was assessed by *t*-test and the ANOVA test, followed by the Bonferroni post-hoc test. The statistical analysis of kinetic data was performed following the “GraphPad guide to comparing dose–response or kinetic curves” [17]. For each kinetic curve obtained in the experimental procedures, the Area Under the Curve (AUC) was calculated to summarize the curve into a single value. Then, statistical analysis was performed on the AUC values of each experimental group using the unpaired *t*-test. The *p*-values were considered significant at *p* ≤ 0.05. Each experiment was repeated three times under the same conditions.

## 3. Results

### 3.1. Isolation of Microorganisms from Different Skin Sites

Out of the 247 strains isolated from the swabs taken from sebaceous, dry, and wet skin sites, 110, 10, and 110 grew on Mannitol Salt Agar, TSA, and Yeast Extract Sodium Lactate agar plates, respectively; according to these results, we hypothesize that they probably belong to the genera *Staphylococcus* spp. and *Propionibacterium* spp. In contrast, only 17 strains grew on de Man–Rogosa–Sharpe agar plates; therefore, we hypothesize that they are lactic acid bacteria.

### 3.2. Detection of Antimicrobial Activity

Of all isolates examined by the deferred antagonism method, only 10% (25 out of 247 strains) showed antimicrobial activity against one or more taxonomically related microorganisms. Among them, CPAY1 CFS displayed the highest antimicrobial activity against the closely related microorganisms used as indicators by the agar well diffusion assay, as shown in Table 1.

These results were confirmed by the agar well diffusion assay, carried out using the CPAY1 CFS that showed a wide inhibitory spectrum against all the microbial indicators, with the strongest antagonistic activity against *S. aureus* strains (Figure 1a). Also of significant interest was the inhibition of other pathogens or opportunists, such as *E. faecalis*, *S. Agalactiae*, and in particular, *C. albicans* and *C. parapsilosis* (Figure 1b). Antimicrobial activity towards Gram-negative bacteria was observed only against *Escherichia coli* ATCC 25922.

### 3.3. Kinetics of Growth and Biosynthesis

*B. siamensis* started to produce antimicrobial compounds in the CFS (approximately 20 AU/mL) after 4 h of culture, during the early log growth phase, when the biomass absorbance was 0.08. The production of the antimicrobial compounds in the CPAY1 CFS was maximal (80 AU/mL) at 14 h of culture, at an absorbance of 1.8. The antimicrobial compounds titer remained constant (absorbance of 1.8) until the end of the observations (24 h).

### 3.4. Effect of Heat Stability, and Protease and pH Sensitivity

The antimicrobial activity of the CPAY1 CFS was stable after treatment at 121 °C for 15 min. The antimicrobial activity was inhibited after treatment with proteolytic enzymes, indicating that the antimicrobial substance might be proteinaceous. Activity was maintained over a wide range of pHs (Figure 2), during storage at room temperature and after 6 months at +4 °C.

### 3.5. Time–Kill Studies

CPAY1 CFS showed activity against viable cells of all the tested strains after 8 h of incubation. It was active against all strains of *E. faecalis* (*p* < 0.01) (Figure 3), while it showed the best activity against *S. aureus* ATCC 6538 (*p* = 0.0016) as compared to MRSA (*p* = 0.0038 and *p* = 0.0022 for *S. aureus* MRSA M1 and MRSA H1, respectively) (Figure 4). As for *E. coli* ATCC 25922, the CPAY1 CFS caused a remarkable reduction in viable cells, as early as after 10 h of incubation. The number of viable *E. coli* cells, when incubated with CPAY1 CFS, kept decreasing until the end of the experiment (*p* = 0.0003) (Figure 5). Finally, a reduced CPAY1 CFS activity emerged against *S. agalactiae* V1 up to the 19th hour of incubation, but it increased again up to the end of the experiment (*p* = 0.0028) (Figure 3d).

### 3.6. Anti-Biofilm Activity

The activity of CPAY1 CFS against 24 h-old biofilms produced by all the microbial strains tested is shown in Figure 6a–d. CPAY1 CFS was shown to be effective against biofilm produced by all *S. aureus* strains tested, with a reduction of 70.45% (*p* = 0.001) for *S. aureus* ATCC 6538, 64.33% (*p* = 0.0004) for *S. aureus* MRSA M1, and 64.00%, (*p* = 0.0042) for *S. aureus* MRSA H1 (Figure 6a). A remarkable reduction in biofilm was observed for *E. faecalis* ATCC 29212 (57%) (*p* < 0.0001) and for all VRE *E. faecalis* strains, with reductions of 42.57% (*p* = 0.0030) for *E. faecalis* A, 49.68% (*p* = 0.0026) for *E*. *faecalis* B, and 43.89% (*p* = 0.0144) for *E. faecalis* C (Figure 6b). Finally, CPAY1 CFS was shown to reduce the biofilm of *E. coli* ATCC 25922 and *S. agalactiae* V1 strains, with a reduction of 55% (*p* = 0.0004) and 38.37% (*p* = 0.0031), respectively (Figure 6c and 6d).

### 3.7. The 16S rRNA Sequencing

According to the results of the antimicrobial screening, we performed a phenotypic identification and 16S rDNA sequencing. The results of this analysis allowed us to classify this strain as belonging to the species *B. siamensis*; we named this strain CPAY1.

### 3.8. Antibiotic Resistance

According to the EFSA breakpoints [14], the CPAY1 strain was assessed for its susceptibility to gentamicin, clindamycin, erythromycin, kanamycin, tetracycline, streptomycin, and vancomycin. The results of the antibiogram showed that *B. siamensis* CPAY1 was susceptible to all the tested antibiotics.

### 3.9. Plasmid Isolation

Figure 7 shows the plasmid profile of the *B. siamensis* CPAY1. The presence of plasmids could not be observed and only the chromosomal band could be detected. Therefore, we hypothesized a chromosomal localization for the gene coding for the bioactive compound(s).

## 4. Discussion

For many years, the human skin has been the subject of studies that have highlighted the presence of a microbiota lining the body surfaces in contact with the external environment. Recent investigations have revealed that resident microbes play a fundamental role as guardians of human health. The skin microbiota, in particular, takes part in protection against infections by interacting with the immune system in humans [18,19]. Indeed, a dysbiosis that involves alterations of the skin microbiota, such as expansion of some microbes compared to others, can lead to the development of autoimmune disorders, inflammatory diseases, and allergic reactions, with a subsequent inflammatory response [20]. Up until a few years ago, antibiotic therapy was the only approach available against skin diseases that were totally or partly linked to microbial colonization. The drawback of such an approach is that the antibiotic therapy alters the balance of the skin microbiota, and while it limits the harmful bacteria, it also kills useful microbes [21]. In addition, because of the growing resistance to the most common antibiotics, they are becoming useless when fighting many microbial pathogens. In recent years, the discovery and characterization of small antimicrobial peptides (AMPs) produced by bacteria have aroused a lot of interest from researchers, who focus on the study of novel antimicrobial compounds as an alternative to address the worrying phenomenon of antibiotic resistance, which is rapidly becoming a global problem. Consequently, the identification of AMPs and their characterization has begun to attract increasing attention [22].

Different bacteria of the genus *Bacillus* have been shown to produce several antimicrobial molecules, such as bacteriocins, glycopeptides, lipopeptides, and cyclic peptides [23,24]. In this study, we report a preliminary antimicrobial characterization of the CFS from *Bacillus siamensis* CPAY1. Several studies demonstrate that *B. siamensis* is a plant-associate species, and it can be found mainly in the rhizosphere of various plants [25,26]. Therefore, the isolation of *B. siamensis* on human skin may indicate that this species may occur as part of the transient skin microbiota and it is capable of colonizing healthy human skin, which is perceived as a favorable environment. From an activity spectrum analysis, CPAY1 has been shown to be active towards several clinical isolates, some of them with a high spectrum of antibiotic resistance. The CPAY1 CFS exhibits strong inhibitory activity against *S. aureus* ATCC 6538, methicillin-resistant *S. aureus* (MRSA), and vancomycin-resistant *E. faecalis* (VRE), but also against some *Candida* strains. MRSA strains have become a major problem worldwide, causing a variety of nosocomial and community-acquired infections [27], such as vancomycin-resistant enterococci causing multiple surgical wound infections [28]. The current study suggests that CPAY1 CFS can be used as an alternative to conventional therapeutic agents or as adjunctive therapy against antibiotic-resistant bacteria, including MRSA and VRE

The antimicrobial activity of the CPAY1 CFS has been retained after treatment at 121 °C for 15 min, but it has been inhibited after treatment with proteolytic enzymes, which suggests a proteinaceous nature of CPAY1 CFS. Indeed, the literature reports several examples of heat-stable antimicrobial substances of a protein nature, such as class II bacteriocins [29,30,31,32]. However, further investigations, such as purification and chemical characterization of the active agent will be carried out in order to evaluate its chemical nature.

Both *S. aureus* and *Candida* strains are opportunistic pathogens that can co-infect the human body, causing urinary tract and skin infections also associated with biofilm production. The high prevalence of drug resistance of *S. aureus*, *C. albicans,* and *C. parapsilosis* makes it even more difficult to eradicate the biofilm, where resistance to the respective antimicrobial agents is already high [33].

High levels of biofilm production by multidrug-resistant organisms (MDROs), primarily responsible for hospital-acquired infections, have been reported in the literature. Nevertheless, biofilm production is not routinely studied in clinical microbiological assays, that evaluate mostly the planktonic microbial forms [34]. Therefore, to evaluate the biofilm impairment capability represents a necessary prerequisite for the development of novel strategies against pathogenic and/or opportunistic microorganisms. Indeed, it was demonstrated that CPAY1 CFS was not only able to prevent the growth of most of the indicator species listed in Table 1, but also to impair a 24 h-old preformed biofilm by several opportunistic pathogens. This is a particularly important result since the biofilm appears to be the main cause of the delay in wound healing [35].

Previous studies had already evaluated the antibacterial activity of compounds produced by *B. siamensis.* In particular, Heo et al. studied a compound produced by *B. siamensis* isolated from Korean fermented vegetable kimchi that has shown antibacterial and anti-biofilm activity against food-borne pathogens and pathogenic bacteria (*Pseudomonas aeruginosa*, *E. coli*, and *S. aureus*) [36]. Tarek et al. have investigated an enzyme protease from *Bacillus siamensis* that is highly stable and effective over a wide range of pHs and temperatures [37]. Xu et al. have identified cyclic lipopeptides from *B. siamensis* that have been employed to suppress the growth of various multidrug-resistant aquatic bacterial pathogens [38]. The CFS of the *B. siamensis* CPAY1 strain described here shows similarly encouraging results and it warrants further investigations to identify and characterize its components. A further requirement for a CFS with antimicrobial activity is the lack of antibiotic resistance of the producer strain because the presence of transferable antibiotic-resistance genes would be at risk of being transferred from probiotic strains to the commensal microbiota in vivo [39]. In this study, *B. siamensis* CPAY1 has proved to be sensitive to all the antibiotics tested, therefore avoiding the risk of resistance genes transferring to the commensal microbiota of the skin. This further supports the idea that *B. siamensis* CPAY1 or its CFS might be useful aids for the treatment of skin diseases, even in association with the current therapies. In addition, CPAY1 CFS has the potential to be employed in other fields (such as the food industry, agriculture, and animal husbandry), where stability at different pHs, at high temperatures, and at refrigeration temperature is fundamental for maintaining the antimicrobial activity over time.

## 5. Conclusions

The increasing interest in the employment of pre-and probiotics, both in the cosmetic and therapeutic fields, aims to improve the compositional properties of the skin microbiota and to contribute to the regeneration of the skin. The effectiveness of probiotics at the intestinal level has been acknowledged for long time. Only recently has there been an attempt to also study probiotics’ effectiveness on the skin microbiota. Furthermore, these living bacteria can produce antimicrobial peptides and improve the body defenses. These results can be obtained by promoting the growth of beneficial microorganisms, as well as counteracting the colonization of pathogens. In the present study, we have performed a partial characterization of the CFS obtained from the *B. siamensis* CPAY1 strain. This CFS has shown antimicrobial and anti-biofilm activity against several microbial pathogens. Furthermore, the absence of virulence factors and the lack of antibiotic resistance warrants further in-depth studies aimed at defining and characterizing the antimicrobial molecules occurring in CPAY1 CFS. The purification, chemical characterization, and cytotoxicity evaluation of its bioactive compounds will provide more precise indications for the employment this *B. siamensis* strain as a probiotic and/or its CFS as a postbiotic, as a support to current antimicrobial therapy.

## Figures and Tables

**Figure 1 microorganisms-12-00718-f001:**
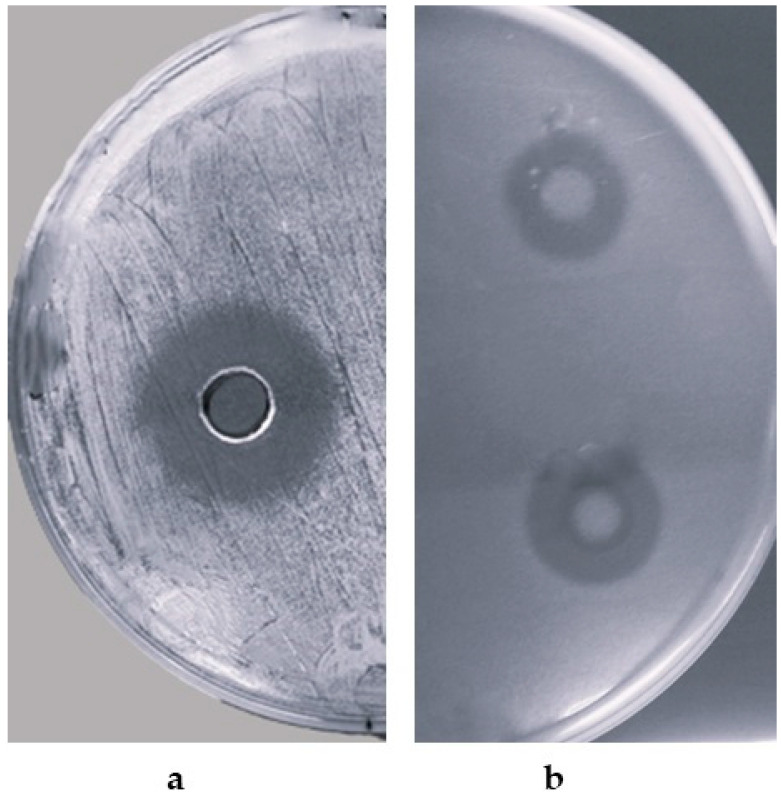
Antimicrobial activity of CPAY1 CFS against *S. aureus* ATCC 6538 (**a**) and *C. albicans* ATCC 10231 (**b**), detected by agar well diffusion assay.

**Figure 2 microorganisms-12-00718-f002:**
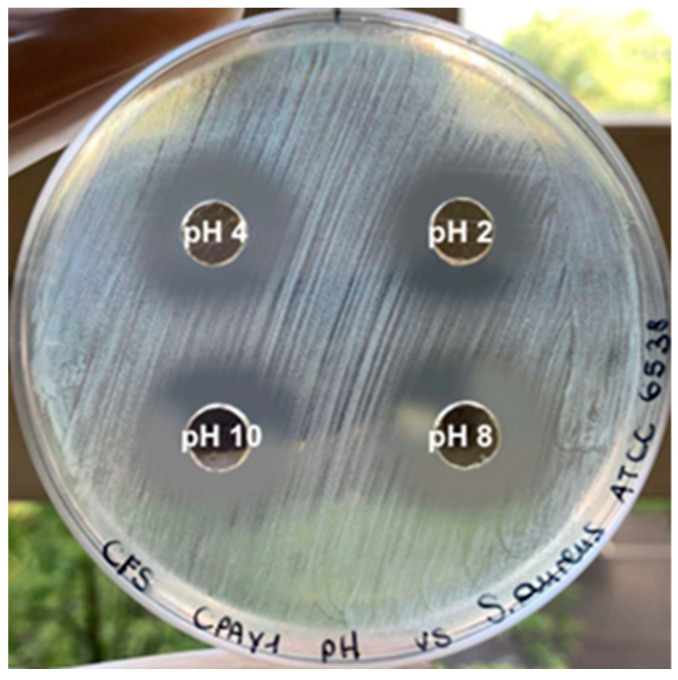
Antimicrobial activity of CPAY1 CFS at different pHs against *S. aureus* ATCC 6538.

**Figure 3 microorganisms-12-00718-f003:**
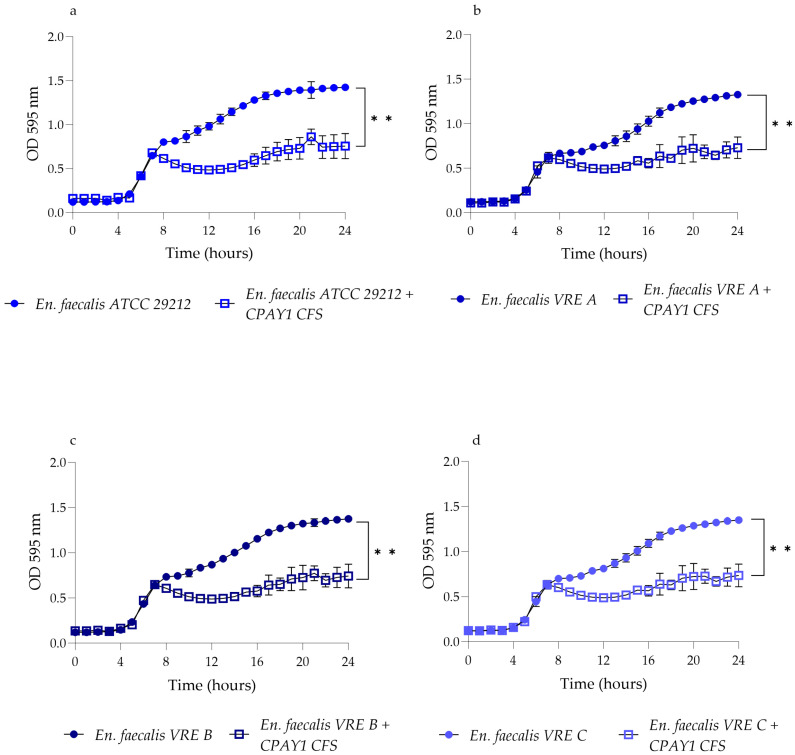
Time–kill studies of CPAY1 CFS against (**a**) *E. faecalis* ATCC 29212, (**b**) *E. faecalis* VRE A, (**c**) *E. faecalis* VRE B, and (**d**) *E. faecalis* VRE C viable cells. Values of *p* < 0.01 (**) was considered significant by *t*-test and ANOVA with Bonferroni correction; ns means no significant. Results were expressed as mean ± SD of the three determinations (error bar = S.D.; *n* = 3).

**Figure 4 microorganisms-12-00718-f004:**
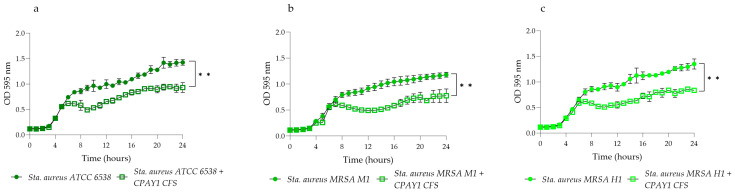
Time–kill studies of CPAY1 CFS against (**a**) *S. aureus* ATCC 6538, (**b**) *S. aureus* M1, and (**c**) *S. aureus* H1 viable cells. Values of *p* < 0.01 (**) was considered significant by the unpaired *t*-test performed on the AUC values of each experimental group. Results were expressed as mean ± SD of the three determinations (error bar = S.D.; *n* = 3).

**Figure 5 microorganisms-12-00718-f005:**
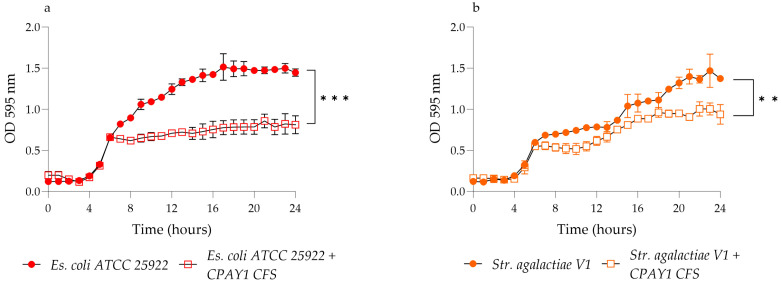
Time–kill studies of CPAY1 CFS against (**a**) *E. coli* ATCC 25922, and (**b**) *S. agalactiae* V1 viable cells. Values of *p* < 0.01 (**) and *p* < 0.001 (***) were considered significant by the unpaired *t*-test performed on the AUC values of each experimental group. Results were expressed as mean ± SD of the three determinations (error bar = S.D.; *n* = 3).

**Figure 6 microorganisms-12-00718-f006:**
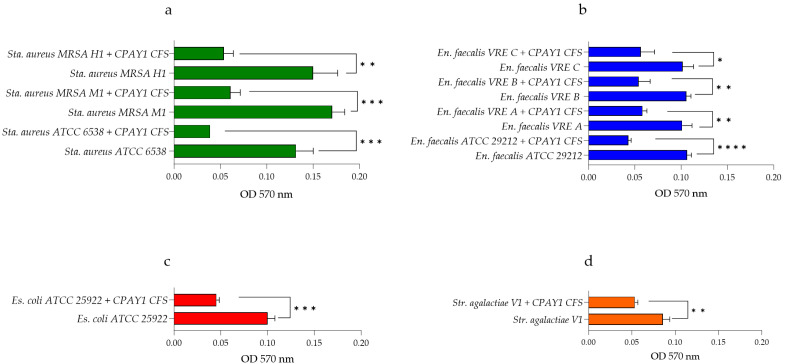
Anti-biofilm activity of CPAY1 CFS against (**a**) *S. aureus* strains, (**b**) *E. faecalis* strains, (**c**) *E. coli* ATCC 25922, and (**d**) *S. agalactiae* V1. Results show the mean optical densities (OD) at 570 nm from three determinations. Values of *p* < 0.05 (*), *p* < 0.01 (**), *p* < 0.001 (***), and *p* < 0.0001 (****) were considered significant by *t*-test and ANOVA with Bonferroni correction. Results were expressed as mean ± SD of the three determinations (error bar = S.D.; *n* = 3).

**Figure 7 microorganisms-12-00718-f007:**
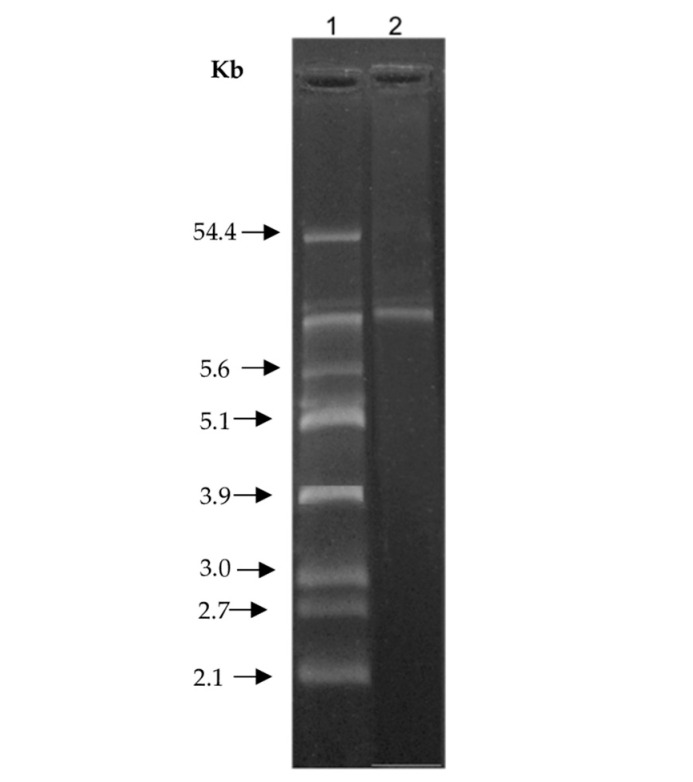
Plasmid profiles of *Bacillus siamensis* CPAY1. Lane 1: molecular size markers prepared from *Escherichia coli* V517 (54.4, 5.6, 5.1, 3.9, 3.0, 2.7, and 2.1 kb); lane 2: *Bacillus. siamensis* CPAY1.

**Table 1 microorganisms-12-00718-t001:** Antimicrobial activity of CPAY1 CFS detected by agar well diffusion assay against different indicator microorganisms.

Indicator Strains	IZ (mm)	Indicator Strains	IZ (mm)
*Staphylococcus aureus* ATCC 6538	+++	*Escherichia coli* ATCC 25922	+
*Staphylococcus aureus* MRSA M1	+++	*Pseudomonas aeruginosa* ATCC 9027	−
*Staphylococcus aureus* MRSA H1	+++	*Klebsiella pneumoniae* ATCC 33495	−
*Staphylococcus epidermidis* IM10	+	*Candida albicans* ATCC 10231	+
*Streptococcus pneumoniae* ATCC 46916	−	*Candida albicans* IM56	++
*Streptococcus agalactiae* V1	+	*Candida albicans* IM 57	++
*Enterococcus faecalis* ATCC 29212	+	*Candida parapsilosis* ATCC 22019	+
*Enterococcus faecalis* VRE A	+	*Candida parapsilosis* IM63	++
*Enterococcus faecalis* VRE B	+	*Candida parapsilosis* IM64	++
*Enterococcus faecalis* VRE C	++	*Candida rugosa* IM67	−

IM label indicates isolates from our collection. −, no inhibition zone (IZ); +, 5 mm < IZ < 10 mm; ++, 10 mm < IZ < 15 mm; +++, IZ > 15 mm.

## Data Availability

The datasets used and analyzed during the current study are available from the corresponding author on reasonable request.

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
