# Peer review of "Cell-Free Supernatant from a Strain of Bacillus siamensis Isolated from the Skin Showed a Broad Spectrum of Antimicrobial Activity"

_microorganisms, 2024, doi:10.3390/microorganisms12040718_

Round 1

Reviewer 1 Report

Comments and Suggestions for Authors

This study deals with the evaluation of antibacterial and antibiofilm activity of cell-free supernatant of a strain of Bacillus siamensis isolated from skin as well as the determination of its sensitivity to chemical-physical parameters, antibiotic-resistance and plasmid presence.

The manuscript is clear and presented in a well-structured manner. The experimental design is appropriate, and the methods are well described. The figures/tables are appropriate and clear. The cited references are mostly recent publications.

            My main concern is the given explanation of the nature of active compounds whereas the crude supernatant was tested without separation of active compounds.

Specific comments:

1. Line 156.  It should be  2.7. 16S rRNA sequencing instead of 2.716. S rRNA sequencing

2. Lines 221-224. The authors noticed that the antimicrobial activity of CPAY1 CFS was not inhibited after treatment with high temperatures (up to 121°C) and proteolytic enzymes. Usually, high temperature and proteolytic enzymes degrade proteins. Maybe, in this case, the CFS was not proteinaceous. Please, give more explanations for the statement that the CFS might be proteinaceous (line 224). The Discussion section also referred to proteins (AMPs) as target compounds, but this is not confirmed.

3. Lines 256, 259, 262. It should be Figure 6 a-d instead of Figure 4 a-d

4. Line 266, figure 6. It should be (a) S. aureus strains, (b) E. faecalis strains instead of (a) E. faecalis strains, (b) S. aureus strains

5. Lines 281, 286. It should be Figure 7 instead of Figure 8

6. Lines 334-337 In the study by Tarek et al., an enzyme, protease from Bacillus siamensis was investigated. In this study, the authors, as potential target compounds, mentioned small antimicrobial peptides (AMPs) (Lines 302-306). Please, be more precise in the discussion of the results.

7. Line 336. The reference 31 is not properly cited, it should be after the statement “Tarek et al. demonstrated that such compound is highly stable and effective over a wide range of pH and temperatures.”

8. Line 339 Delete cited reference 31. It was referred to the previous sentence (lines 336-337).

Author Response

Thank you very much for taking the time to review this manuscript. Please find the detailed responses below and the corresponding revisions/corrections highlighted/in track changes yellow color  in the resubmitted files

Reviewer 2 Report

Comments and Suggestions for Authors

While the methodology section adequately describes the isolation and characterization process of bacterial strains from the skin, there are notable shortcomings. The absence of information regarding the selection criteria for indicator strains and the lack of detail on the preliminary investigation of biofilm-forming abilities of the same strains are notable gaps. Additionally, clarity is needed regarding positive and negative controls in the well agar diffusion method.

The discussion section effectively contextualizes the findings within the broader scope of skin microbiota research and the current challenges posed by antibiotic resistance. However, there is some redundancy with the introduction, and more emphasis on interpreting the specific results in light of existing knowledge would strengthen this section.

Author Response

(The authors gave the same response as above.)

Reviewer 3 Report

Comments and Suggestions for Authors

The work is devoted to a current topic - the search for new natural agents for medical use that overcome the antibiotic resistance of pathogenic microorganisms. The authors searched for potential producers among bacteria isolated from the surface of the human body. This approach suggests that the resulting antimicrobial substance is nontoxic and may be promising for medical use.

Of the 247 isolated strains, a tenth had antibiotic activity. Further work was carried out with the most active CPAY1 strain, which was identified as Bacillus siamensis. Activity assays were performed using Cell Free-Supernatant. The substance(s) produced by this strain exhibited broad-spectrum activity, including against a number of antibiotic-resistant strains of clinical significance. In addition to antibiotic activity, film formation activity has been shown on different strains.

In the Discussion section, the authors cite publications on Bacillus siamensis and conclude that since representatives of this species were previously described as associated with plants, it can be considered as a transient species of the human skin microbiota.

The work was carried out at the appropriate methodological level.

The text of the manuscript contains the following questions:

1. "The skin epidermis, with its surface of about 25 m2, is one of the largest epithelial surfaces of the human body..." Line 56

The average body surface area is 1.6-2.2 m2. Please clarify.

2. The generic name is given in full in the text at the first mention, then abbreviations must be entered. Abbreviations should be different for Staphylococcus and Streptococcus, Enterococcus and Escherichia. (Table 1, Figures 3-5).

3. Lines 222-226.

"The antimicrobial activity did not change after treatment with proteolytic enzymes indicating that the antimicrobial substance might be proteinaceous."

 This result does not categorically confirm that the active agent is a protein. Although it does not rule out that this is a protein resistant to these enzymes.

Author Response

(The authors gave the same response as above.)

Round 2

Reviewer 1 Report

Comments and Suggestions for Authors

The authors have made necessary corrections and improved the quality of the paper. The paper, in its present form, is suitable for publishing. 

Author Response

We would like to thank reviewer for kind comments on the manuscript.

Reviewer 2 Report

Comments and Suggestions for Authors

I agree with the revisions made to the text, highlighted in yellow. However, it's important to note that according to microbiological nomenclature, the genus name of a microorganism should be abbreviated with only the first letter capitalized when using abbreviations.

Regarding my suggestion for improvement, including a picture of the negative controls used at pH 2, 4, and so on, would indeed enhance the quality of the study by providing a visual representation of the experimental setup and controls.

Author Response

I agree with the revisions made to the text, highlighted in blue.
